# Crystal Plasticity Simulation of Yield Loci Evolution of SUS304 Foil

**DOI:** 10.3390/ma15031140

**Published:** 2022-02-01

**Authors:** Mingliang Men, Bao Meng

**Affiliations:** School of Mechanical Engineering and Automation, Beihang University, Beijing 100191, China; Bright_men@163.com

**Keywords:** yield loci, crystal plasticity, grain size, crystal orientation

## Abstract

The deformation process of metal foils is usually under a complex stress status, and the size effect has an obvious influence on the microforming process. To study the effect of grain orientation and grain size distribution on the yield loci evolution of SUS304 stainless steel foils, three representative volume element (RVE) models were built based on the open source tools NEPER and MTEX. In addition, the yield loci with different grain sizes are obtained by simulation with Duisseldorf Advanced Material Simulation Kit (DAMASK) under different proportional loading conditions. The initial yield loci show a remarkable difference in shape and size, mainly caused by the distinct texture characteristics. By comparing the crystal plasticity simulation with the experimental results, the model with normal grain size distribution and initial texture based on Electron Back-scattered Diffraction (EBSD) data can more accurately describe the influence of the size effect on the shape and size of yield loci, which is the result of the interaction of grain size distribution and texture. However, the enhancement of grain deformation coordination will weaken the impact of the size effect on yield loci shape if the grain size distribution is more uniform.

## 1. Introduction

In recent years, as the basis for product miniaturization and integration, metal foils and their components are widely used in electronic communications, aerospace, new energy vehicles, and biomedical fields. Due to the advantages of high productivity, less material loss, and excellent mechanical properties, microforming has been widely used in the fabrication of microcomponents in various fields [1,2,3,4]. Metal foil is composed of a large number of grains and may form texture after multi-pass rolling or heat treatment. In the microforming process, the size of microscopic features is impossible to ignore relative to the specimen size. Although there is coordinated deformation among grains, the morphology of the microstructure and deformation behavior have a large impact on the overall behavior of the material. It will show phenomena different from the macroscopic behavior [5,6] even under simple loading conditions, such as uniaxial tension, biaxial tension, and forming limit. Since this size effect has an important influence on the plastic deformation behavior of metal foil, it is necessary to explore the size effect on the yield behavior of metal foil.

Under the assumption of continuum mechanics, macro-scale theoretical research is carried out to obtain the mechanical response of metal sheets through basic mechanical experiments. Afterward, investigators can acquire the physical laws by summarizing these experimental facts to establish the constitutive model. Plastic models based on yield functions are widely used in finite element analysis. However, these macroscopic plastic models do not contain any microscopic characteristic parameters, which cannot explain the size effect of materials at the micron scale. Therefore, they are no longer suitable for accurate analysis of microforming processes. Crystal plasticity (CP) is a physics-based constitutive model that leverages the material’s microstructure to determine plastic slip within polycrystalline metals. Physical models and finite element simulations based on CP theory play an important role in understanding the yield and anisotropy of metals. The Taylor–Bishop–Hill (TBH) model was used to generate the analytical expressions for yield surfaces of anisotropic polycrystalline materials [7]. Shi et al. [8] explained the size effect on the evolution of subsequent yield loci in terms of texture and its evolution. It should be worth noting that although the TBH model satisfies the compatibility condition between grains with different orientations, it cannot satisfy the stress equilibrium condition. Nevertheless, the full-field CP model based on polycrystalline plasticity theory can realize both [9]. The crystal plasticity finite element method (CPFEM) has been widely used at the meso-level, which can simulate the macro and micro mechanical response, deformation behavior, and microstructure evolution of metal materials under complex physical boundary conditions [10]. 

Many researchers have investigated the yield behavior with CPFEM. Zhang et al. [11,12] established a RVE model of polycrystals, improved the classical CPFEM [13], simulated and analyzed the mechanical behavior of materials, and introduced nonlinear kinetic strengthening parameters on this basis, which optimized the simulation results [12,14]. Then, different test methods were used to study the subsequent yield surfaces of pure copper samples under different loading paths [15,16], and the crystal simulation results were in good agreement with the experimental results. Lu et al. [17] introduced the anisotropic strain hardening back stress into the crystal plasticity theory and demonstrated the rationality of the crystal plasticity model in describing the subsequent yield surface evolution of polycrystalline aluminum at the mesoscale under a complicated pre-cyclic loading path. It was found that the size and shape of the subsequent yield surfaces are very sensitive to the selected bias strain and the direction of pre-cyclic loading, and the anisotropic hardening of the yield surfaces is related to the crystal microstructure and the inhomogeneous deformation caused by crystal slip. Hu et al. [18] studied the cyclic tension–compression yield of polycrystalline aluminum by using experiments and crystal plasticity simulations, and they analyzed in detail the effects of different unloading positions, loading directions, and yield definitions on subsequent yield surfaces. Toth et al. [19] pointed out that the whole macroscopic Bauschinger effect was caused by the first yielding of soft-oriented grains in the polycrystal model. The combination of the spectral method based on fast Fourier transform (FFT) and full-field CP model (CPFFT/CPSM) is more efficient in calculating and solving boundary value problems (BVPs) [9,20]. Diehl et al. [20] provided a virtual laboratory [9] to study the anisotropic yield behavior of polycrystalline materials by using high-resolution crystal plasticity simulations. The evolution of the anisotropic yield function was predicted by combining the large-scale forming simulation with CPFEM, and a multiscale model of metal forming was established. The mechanical properties of 2090-T3 aluminum alloy were investigated, and the proposed method was integrated into DAMASK [21]. Cai et al. [22] established polycrystalline RVE models for dual-phase (DP) steel and 6016-T4P based on CPFEM/CPSM and then carried out biaxial tensile simulation. The yield loci obtained by the simulation were in good agreement with the experimental results. Liu et al. [23] also used CPFFT to predict the anisotropic mechanical properties of aluminum alloy sheets under uniaxial and multiaxial stress states.

At present, there are few studies on the yield loci of metal foil using crystal plasticity. In this paper, SUS304 stainless steel foil is taken as the research object, and three kinds of polycrystal models with different grain sizes are established. Combined with the crystal plastic constitutive theory and crystal plasticity open-source tool DAMASK, simulations under different linear loading conditions were performed to obtain the yield loci. The effects of different crystal orientations and grain size distributions on the initial yield loci of foil are investigated to establish the relation between macroscopic yield and microstructure.

## 2. Materials and Methods

### 2.1. Materials

The material used in this paper is SUS304 stainless steel foil with a thickness of 120 μm. In order to study the influence of different grain sizes and crystal orientations on the initial yield loci of stainless-steel foil, heat treatment was carried out to obtain different properties. The heat treatment process is shown in Table 1 [8], and the corresponding microstructure is shown in Figure 1 [8]. In this paper, the ratio of the foil thickness *t*_0_ to the average grain size *d*_0_ is the thickness grain number, namely, the size factor λ. It can be seen from the figure that the texture phenomenon of SUS304 foil with *λ* = 3.89 is not obvious, and the grain orientation is in the state of random distribution. When *λ* = 1.97, the grain orientation in the foil is dominated by (110)[001]. When *λ* = 1.75, the number of grains with $\left(111)[1\bar{3}2\right]$ orientation is large.

### 2.2. Experimental Process

The shapes and dimensions of specimens for uniaxial and biaxial tests are presented in Figure 2 [8]. To obtain the mechanical properties and flow stress–strain curves of SUS304 foil, the uniaxial tensile tests were carried out on the Materials Test Systems (MTS) universal test machine (MTS Systems, Eden Prairie, MN, USA) with a strain rate of 0.0017 s^−1^. Because of the advantages of detecting full-field strain in foil deformation, the digital image correlation method (DIC, Beihang University, Beijing, China) is used to replace the traditional extensometer (MTS Systems, Eden Prairie, MN, USA). Therefore, by analyzing the deformation behavior of uniaxial specimens, the true stress–true plastic strain curves of SUS304 can be obtained as shown in Figure 3, which are used to calibrate parameters in subsequent simulations. The two directions of the cruciform specimens, that is, rooling direction (RD) and transverse direction (TD) were directly loaded in different proportions, referring to *F_x_*:*F_y_* = 4:0, 4:1, 4:2, 4:3, 4:4, 3:4, 2:4, 1:4, and 0:4. Based on the biaxial loading test platform developed by Beihang University (Beijing, China) for metal foil and optimized cross-shaped samples, the biaxial tensile tests were carried out under different loading ratios. Similarly, the DIC method (Beihang University, Beijing, China) was employed to measure the strains of two directions [8].

### 2.3. Theory of Crystal Plasticity

There have been many reports on the finite element model of the single crystal plasticity constitutive relation based on the cubic crystal structure. The crystal plasticity finite element model adopted in this paper is the classical phenomenological law in DAMASK [21] as the constitutive model. If there is no thermal expansion or crack opening, the deformation of a single crystal can be decomposed into the following two parts: elastic deformation and plastic deformation. The total deformation gradient ***F*** of the crystal can be read as:(1)$$F=F^{e}F^{p}$$
where $F^{e}$ and $F^{p}$ represent the elastic and plastic deformation gradients, respectively.

Elastic deformation satisfies the generalized HOOKE’s law. Plastic deformation can be described in terms of rate:(2)$${\dot{F}}^{p}=L^{p}\cdot F^{p}$$
where $L^{p}$ is the plastic velocity gradient. Suppose that the slip shear strain rate of the *α*th slip system is ${\dot{\gamma }}^{\left(\alpha \right)}$, then $L^{p}$ can be obtained by summing up the contributions of each slip system, which can be expressed as:(3)$$L^{p}=\sum_{\alpha =1}^{N}{\dot{\gamma }}^{\left(\alpha \right)}m^{\left(\alpha \right)}⊗n^{\left(\alpha \right)}$$
where $m^{\left(\alpha \right)}$ and $n^{\left(\alpha \right)}$ are the slip direction and normal vector of the slip plane of the *α*th slip system, respectively, and *N* is the total number of slip systems in the crystal.

The crystal slip obeys typical Schmid’s law for the face-centered cubic lattice types, and the shear stress, i.e., the driving force $\tau^{\left(\alpha \right)}$ on the crystal slip system is:(4)$$\tau^{\left(\alpha \right)}=M_{p}\cdot P_{\mathrm{Schmid}}^{\alpha }=M_{p}\cdot (m^{\left(\alpha \right)}⊗n^{\left(\alpha \right)})$$
where, $M_{p}$ is the Mandel stress, which can be calculated from the second PIOLA–KIRCHHOFF stress. In the rate-dependent crystal plastic constitutive relation, based on Schmid’s law, the rate-dependent viscoplastic model is adopted as the hardening model of crystal materials. ${\dot{\gamma }}^{\left(\alpha \right)}$ can be determined by its corresponding decomposed shear stress $\tau^{\left(\alpha \right)}$:(5)$${\dot{\gamma }}^{\left(\alpha \right)}={\dot{\gamma }}_{0}^{\left(\alpha \right)}{\left(\frac{\tau^{\left(\alpha \right)}}{\tau_{0}^{\left(\alpha \right)}}\right)}^{n}\mathrm{sgn}(\tau^{\left(\alpha \right)})$$
where ${\dot{\gamma }}_{0}^{\left(\alpha \right)}$ is the reference shear strain rate on the *α*th slip system, and in this paper, ${\dot{\gamma }}_{0}=0.001$. *n* is the rate-sensitive index. When n→∞, this equation is close to the description of the rate-independent equation, and here, *n* = 20. sgn(x) is a symbolic function. $\tau_{0}^{\left(\alpha \right)}$ represents the strain hardening strength of the slip system at a certain moment, that is, the slip resistance, and its hardening rate can be expressed as:(6)$${\dot{\tau }}_{0}^{\left(\alpha \right)}=\sum_{\beta }h_{\alpha \beta }\left|{\dot{\gamma }}^{\left(\beta \right)}\right|$$
where $h_{\alpha \beta }$ is the self-hardening modulus (α=β) and latent hardening modulus (α≠β). The strain hardening law of the slip system, namely its hardening modulus, can be expressed by the following equation [24]:(7)$$h_{\alpha \beta }=q\cdot h_{0}\cdot {\left(1-\frac{\tau_{0}^{\left(\alpha \right)}}{\tau_{sat}^{\left(\alpha \right)}}\right)}^{a}$$
where, $h_{0}$ is the initial hardening modulus at yield, and $\tau_{sat}^{\left(\alpha \right)}$ is the saturated slip resistance. *a* is the hardening exponent, and in this paper, *a* = 2.25 is taken. *q* is the coefficient describing the relative strength of self-hardening and latent hardening, which is a constant. Generally, 1 ≤ *q* ≤ 1.4, and *q* = 1 in this paper. More details can be found in the Ref. [21].

### 2.4. Crystal Plasticity Simulation

In this paper, the open-source software NEPER [25] was used to build geometrical polycrystal representative volume element (RVE) models. The average grain size was consistent with the experimental results. Generally, to accurately represent the initial microstructure of foils, adequate grain orientations should be extracted from the orientation density function (ODF) determined by EBSD data, and assigned to each element of the RVE model. Hielscher et al. [26] developed a pole figure inversion approach to give a least-squares (LS) estimator of the unknown ODF. A given number of grain orientations can be uniquely generated with this method, which has been implemented as part of the open-source crystallographic toolbox MTEX [27]. Li et al. [28] used MTEX to discretize ODF to obtain a given amount of grain orientations, which were then assigned to the RVE model of 7075 aluminum alloy. The process of building the complete model and simulating is shown in Figure 4. 

The models are divided into the following: model 1, model 2 and model 3. Model 1 has random orientation and normally distributed grain size. The orientation of model 2 is given based on the EBSD data measured by the experiment, and the grain size distribution obeys the normal distribution. The crystal orientation of model 3 is given according to the experimental data, and the grain size distribution is uniform. The crystal texture and grain size distribution are shown in Figure 5, where *λ* = 3.89, *λ* = 1.97, and *λ* = 1.75 from top to bottom in each model.

The main phase of 304 stainless steel foil is the face-centered cubic (FCC) austenite phase. Because the FCC material only has one set of slip systems, its material parameters calibration is relatively simple. A more common approach is to fit the macroscopic stress–strain curve and texture evolution of polycrystalline materials. In this paper, the model parameters were calibrated according to the trial-and-error method [12]. Figure 6 shows the comparison between the simulation and experimental results. The material parameters corresponding to these results are the calibration values, which will be used in the subsequent simulation under different linear loading conditions to obtain the yield loci. Among them, the values of elastic constants were obtained from Reference [29], *C*_11_ = 209 GPa, *C*_12_ = 133 GPa, and *C*_44_ = 121 GPa. Other parameters for the different models are shown in Table 2.

## 3. Results and Discussion

### 3.1. Experimantal Results

Based on the principle of equal plastic work per unit volume, the plastic work contour corresponding to 0.2% equivalent plastic strain is the initial yield locus, as shown in Figure 7. As the size factor decreases, not only the yield locus gradually shrinks inward, but also the shape of the yield locus also changes. The experimental yield loci of the foils corresponding to size factors *λ* of 3.89, 1.97, and 1.75 are close to elliptical, square, and polygon shapes, respectively. This characteristic is consistent with previous studies [5,30] and can be explained by the physical definition of yield stress at the microscopic level. Furthermore, for metal foils with size factors of 1.97 and 1.75, the thickness grain number and average grain size were relatively close, but the shapes of the yield loci were quite different. Combined with the polar diagram of Figure 1, it can be seen that the two have different preferred orientations, that is, different texture types. The max polar density increases from 3.408 for λ = 1.75 to 9.492 for λ = 1.97, which indicates that the textural phenomenon of foil is significantly enhanced. In other words, this distinction in yield loci may be caused by differences in texture within metal foil.

### 3.2. Simulation Results

Similarly to the measuring method of the experimental yield points, the comparison between simulated yield loci (Sim.) based on different models and experimental results (Exp.) is shown in Figure 8.

The simulation results of different grain size distributions and grain orientations have different deviations. To quantitatively evaluate the accuracy of the different models to the test yield loci, an error function is introduced, namely:(8)$$\begin{array}{l} E=\frac{1}{n}\sum_{i=1}^{n}\frac{d_{i}}{\sqrt{{\left(\sigma_{\exp1}^{i}\right)}^{2}+(\sigma_{\exp2}^{i}{)}^{2}}}\times 100\%\\ d_{i}=\sqrt{{\left(\sigma_{\exp1}^{i}-\sigma_{\mathrm{sim}1}^{i}\right)}^{2}+{\left(\sigma_{\exp2}^{i}-\sigma_{\mathrm{sim}2}^{i}\right)}^{2}} \end{array}$$
where $\sigma_{\exp1}$ and $\sigma_{\mathrm{sim}1}$ are the major principal stress of experimental and simulated results, respectively. $\sigma_{\exp2}$ and $\sigma_{\mathrm{sim}2}$ are the minor principal stress of experimental and simulated results, respectively. $d_{i}$ is the distance between the experimental yield point and the theoretically predicted yield point under the corresponding loading path. *N* represents the number of yield points on each test yield locus. According to Equation (8), the errors between the experimental and theoretical yield loci with different size factors under the three models are calculated, as shown in Figure 9.

### 3.3. Discussion

In model 1, when *λ* = 3.89, the simulation result is in good agreement with the experimental result, and there is only approximately 1% error. With the decrease in the size factor, the deviation of the simulation results gradually increases, especially when the size factor drops from 3.89 to 1.97, and the error increases instantly from 1% to about 8%. For *λ* = 1.97 and 1.75, the yield loci of the two are close to each other except for 0° and 90° directions, and the errors are relatively close to 8%. There is a large deviation in the upper half of the yield loci. This is mainly because under the same random orientation condition, the coordination ability between grains weakens with thickness grain number decreasing, and the shape and size of single grains in the established RVE model have a great influence on the anisotropy behavior, thus affecting the simulation results. 

For model 2, a similar law can be observed. However, model 2 is given true orientations based on EBSD data. By comparing Figure 8a,b, it is obvious that the accuracy of model 2 is better than model 1 when the grain size distribution and average grain size are the same. In the case of a large size factor (*λ* = 3.89), the error of model 2 is higher than that of model 1, which may be because the crystal orientation has greater influence due to the increase in its polar density. However, it is still less than 3%, which can better describe the experimental results. When the size factor decreases, the error increases at a slower speed, and both are smaller than those of model 1. It is the main reason that although both model 1 and model 2 have grain size distribution similar to the actual situation, the latter has initial texture based on EBSD data, which can reflect the real microstructure of the metal foil. That is to say, the simulation results of this model can better capture the shrinkage and deformation of the yield loci. 

The error of model 3 increases first and then decreases with size factor from 3.89 to 1.97 to 1.75 in Figure 9. It can be seen from Figure 8c that the yield loci shapes of model 3 are all closer to an ellipse. Although the shape change of the yield loci is not as apparent as that of the other two models, the error value of λ = 1.75 is lower. This is mainly because the grain distribution in model 3 is more uniform and the coordination between grains in the model is stronger, which weakens the effect of orientation on the shape change of yield loci.

In summary, the high accuracy of model 2 benefits from the consideration of grain size distribution and initial texture close to the real state. Therefore, model 2 can better describe the impact of the size effect on the inward shrinkage and shape change of yield loci, which is the result of the synergy of grain size distribution and orientation texture. Additionally, it is worth noting that the applied crystal plasticity model does not contain parameters considering the size effect. The size effect described in this work is obtained under different parameters, and a higher precision constitutive model with physical significance needs to be further explored.

## 4. Conclusions

This work uses the NEPER and MTEX toolbox to build crystal plasticity models with different grain orientations and size distributions under different size factors. Then, simulations under different proportions of loading are carried out to obtain the initial yield loci. The ability of different models to describe experimental results and the difference of yield loci are analyzed and discussed. The main findings are summarized as follows:(1)With the decrease in the size factor, the yield locus shrinks inward and its shape changes. The size effect has a significant impact on both size and shape of the yield locus. This variation is mainly caused by the different texture characteristics of metal foils.(2)When the average grain size is the same, as the size factor decreases, the error shows an increasing trend. When the grain size distribution is more uniform, the enhancement of grain deformation coordination will weaken the influence of size effect on yield locus shape.(3)When the model is given the real orientation texture and normal grain size distributation, finite element simulation of crystal plasticity can better capture the change of yield locus shape and size.

## Figures and Tables

**Figure 1 materials-15-01140-f001:**
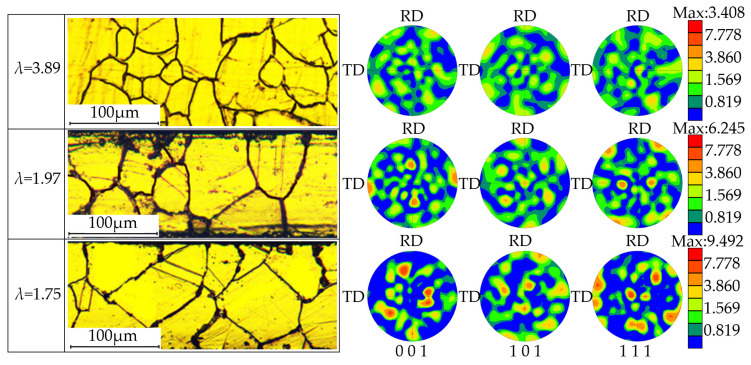
Microstructure of SUS304 foil in different states.

**Figure 2 materials-15-01140-f002:**
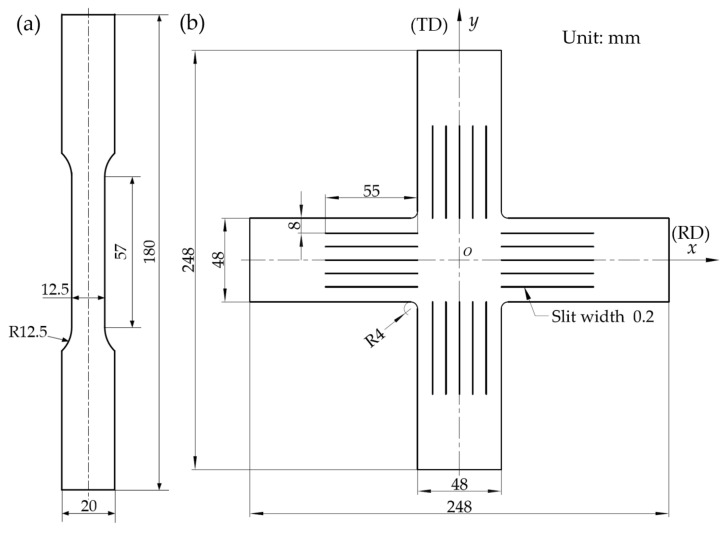
Geometries and dimensions of specimens in: (**a**) uniaxial; (**b**) biaxial tensile tests [8].

**Figure 3 materials-15-01140-f003:**
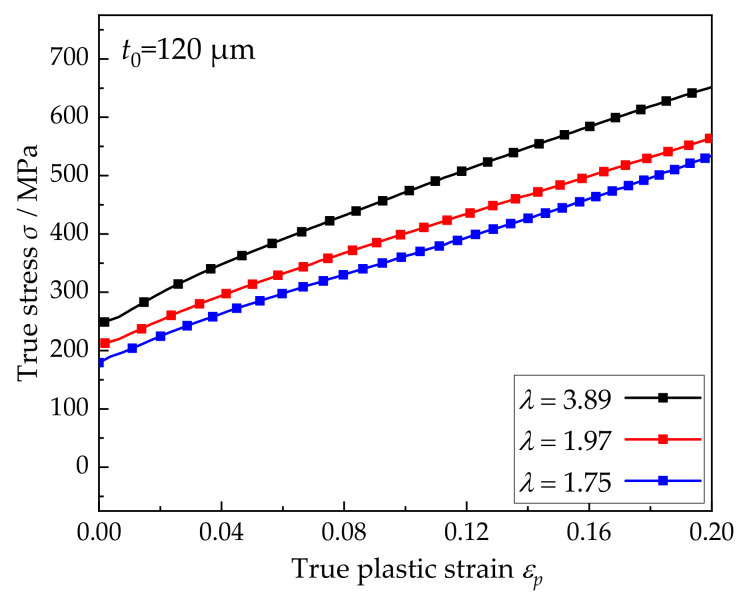
True stress–true plastic strain curves of foils with different grain sizes.

**Figure 4 materials-15-01140-f004:**
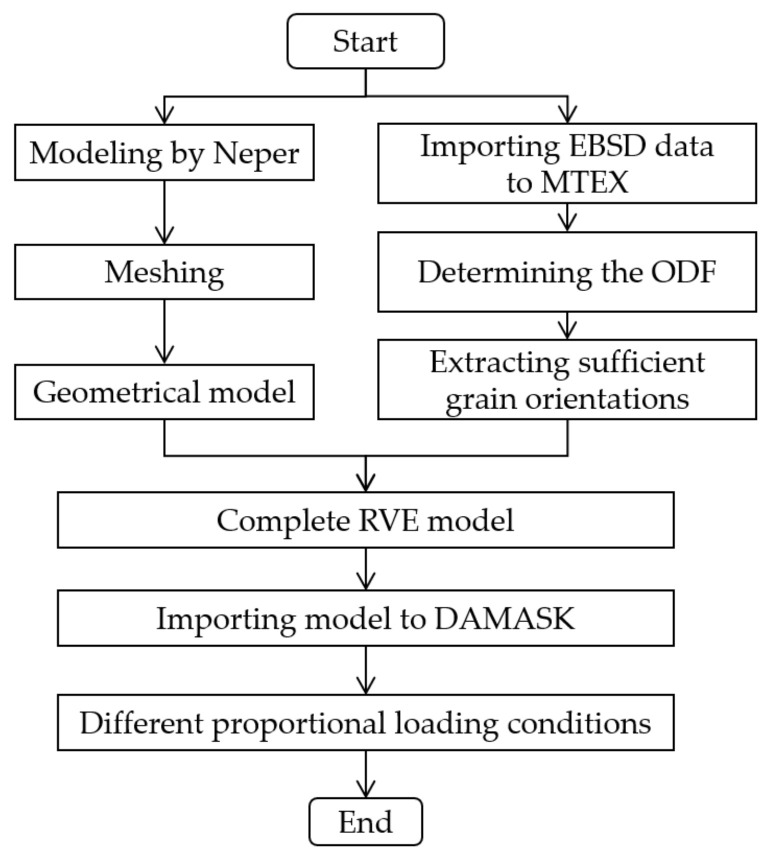
Flow chart of building the complete model and simulation.

**Figure 5 materials-15-01140-f005:**
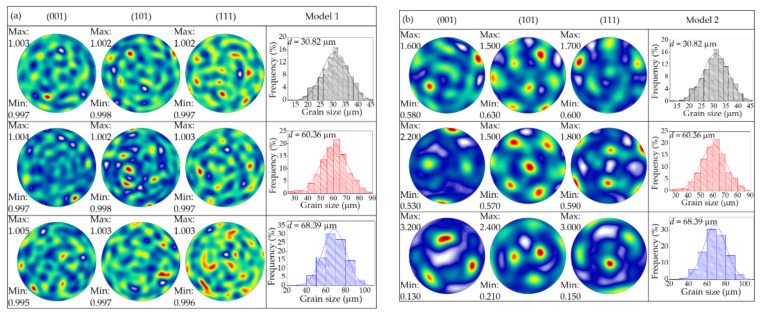
Crystal orientation and grain size distribution: (**a**) model 1; (**b**) model 2; (**c**) model 3.

**Figure 6 materials-15-01140-f006:**
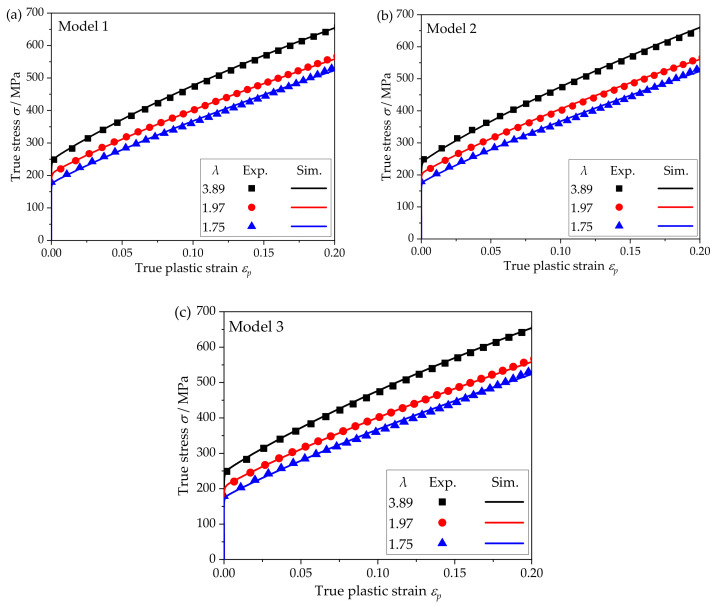
The experimental stress–strain curves of different models and the simulation results: (**a**) model 1; (**b**) model 2; (**c**) model 3.

**Figure 7 materials-15-01140-f007:**
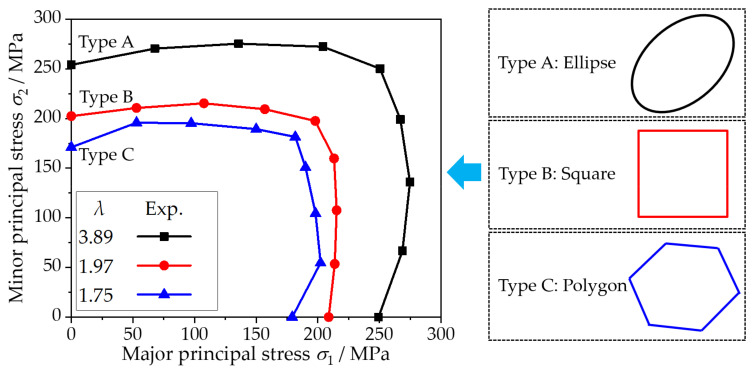
Initial yield loci of metal foils with the thickness of 120 μm and different grain sizes.

**Figure 8 materials-15-01140-f008:**
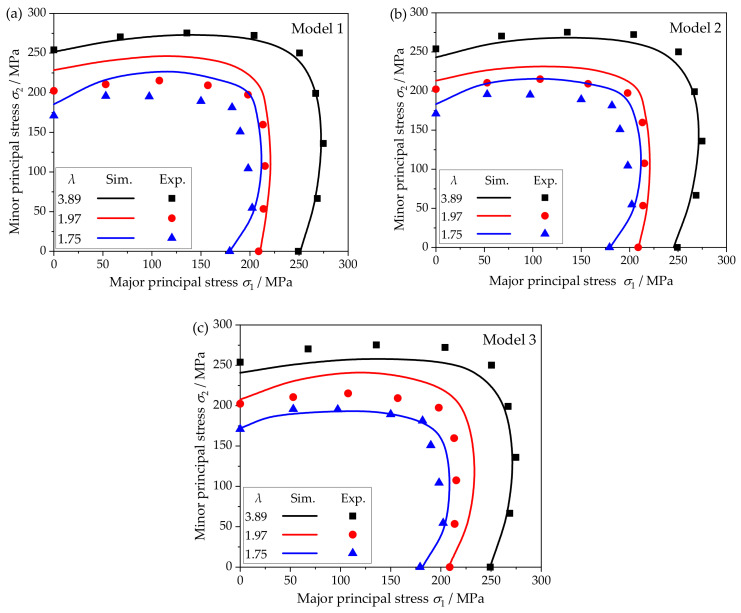
Comparison of simulated and experimental yield loci in different models: (**a**) model 1; (**b**) model 2; (**c**) model 3.

**Figure 9 materials-15-01140-f009:**
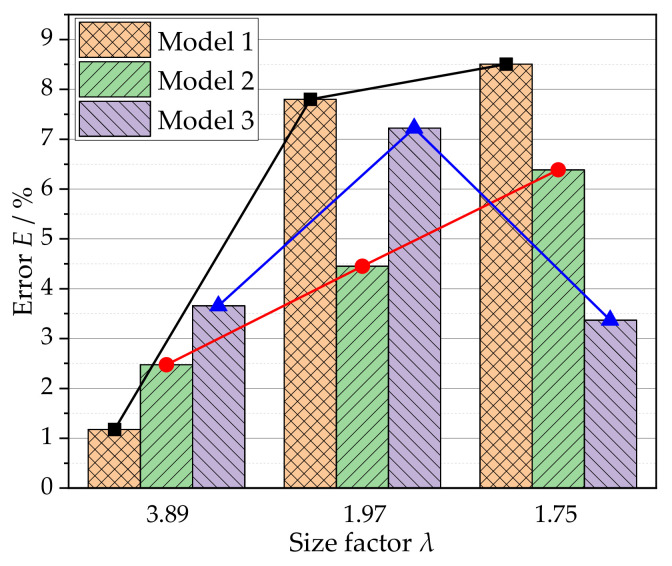
Errors between simulated and experimental yield loci under different size factors.

**Table 1 materials-15-01140-t001:** Annealing parameters of SUS304 foils [8].

Annealing Parameters			Average Grain Size *d*_0_ (μm)	Grain Size Deviation Δ*d*_0_ (μm)	Size Factor *λ* = *t*_0_/*d*_0_
No.	Annealing Temperature (°C)	Holding Time (min)			
1	1000	10	30.81	7.9	3.89
2	1050	30	60.87	10.1	1.97
3	1100	60	68.41	15.6	1.75

**Table 2 materials-15-01140-t002:** Parameters of models with different grain sizes.

Model	*λ*	$h_{0}\;(\mathrm{MPa})$	$\tau_{0}\;(\mathrm{MPa})$	$\tau_{sat}\;(\mathrm{MPa})$
1	3.89	430	96	750
	1.97	380	84	730
	1.75	325	72	900
2	3.89	460	96	750
	1.97	340	76	580
	1.75	325	69.5	880
3	3.89	460	92	600
	1.97	380	82	700
	1.75	290	65	610

## Data Availability

The data presented in this study are available upon reasonable request from the corresponding author.

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
