# Peer review of "Crystal Plasticity Simulation of Yield Loci Evolution of SUS304 Foil"

_materials, 2022, doi:10.3390/ma15031140_

Round 1
Reviewer 1 Report
Since metal foils find a wide applicability in the aerospace industry, in medicine, electronics or in microsystems technology, a modeling study on the orientation of the grains and on their size under the influence of plastic deformation, has a special importance.
In this paper, models of crystalline plasticity were built, with different orientation of the crystallographic grains and the distribution of the grain size, under different size factors, based on the NEPER and MTEX ​​working tools. By comparing the simulation results of 3 models using the EBSD technique, with the experimental results, it was concluded that they are in good agreement when there is the same distribution of grain size. If the orientation of the grains is the same, the simulation of the model with a normal grain distribution proved to be better than those with a uniform grain size distribution.
The paper is well written, well-founded and the discussions explain very clearly the results obtained.
I recommend this paper for publication and congratulate the team of authors.
Author Response
Thanks very much for your affirmation and recommendation of this paper
Reviewer 2 Report
The authors present a CPFEM method and simulation to achieve the evolution of loci for SUS304 foil. Compared with experimental data, the simulation shows a good prediction of texture evolution and yield loci. The topic of the manuscript is of scientific interest. However, a minor revision is recommended as there are several questions/deficiencies to be addressed.
- In this work, the grain size effect is not considered based on either a strain gradient (GND model) or stress gradient model (Hall-Petch). Instead, the effect of grain size is included in a hardening term in eq 11, and the size effect is dependent on several parameters listed in Table 2. How to obtain these parameters was not given in the manuscript. On line 166, the authors mentioned the calibration of these parameters to Ref 23. However, ref 23 shows an investigation of Mg alloy, and details about how to achieve these parameters are also not given.
- In the discussion section, the authors attribute the error of the yield locus to the orientation texture and grain size distribution. On line 205, it claims that when the same orientation texture with the real one is applied, the simulation results can have a better prediction. This can be found by comparing the pole figure of model 1 and 2 directly. However, the pole figure only denotes the intensity or frequency of grain orientation without considering the spatial distribution of the grains. Thus, the local orientation and the grain size (especially for model 1 and 2) would have a synergetic effect on the local slip. So the question comes to how do the authors assign the grain orientation to each grain? And how does the model consider the synergetic effect of grain orientation and grain size?
Reviewer 3 Report
Please correct following major issues:
- There are no size effects in the crystal plasticity model that you use. Different hardening law parameters and initial texture are the cause for different behavior, not intrinsic size effect present in the model. Please correct the manuscript to clearly convey this.
- Please have someone with good English review the paper for grammatical errors and strange and unusual expressions.
- In section 2.2, the authors describe the formulation based on Jaumann stress rate, however this is not what is used in DAMASK. DAMASK constitutive equations are not in rate form nor in the current configuration. In DAMASK second Piola-Kirchhoff stress is defined based on the elastic Green-Lagrange strain. Look in section 6 of Roters et al. (2019) paper that you cite.
- Please improve the description in 2.3 section. Right now it is not understandable. What are different forms? What are modes? What does figure 2 represent?
- Please again be more clear when discussing your results in section 4. It is difficult to follow what you want to convey in section 4.
Minor issues:
Line 35: “Professor Zhang Keshi's team has done a lot of research on crystal plasticity and the evolution of yield loci.” – certainly there are more important people for the field of crystal plasticity and yield loci, emphasizing one researcher over the others, which have done much more is not really fair. Please remove the sentence.
2.1: Please define t_0 and lambda in text.
Figure 1: Please replot the pole figures with the same scale, so that they can be compared more easily.
2.1: Information for speed of 5mm/min is not of importance without the sample size information. Please provide the strain rate, which is more descriptive for examination of material response.
2.2: L is velocity gradient, term velocity deformation gradient is wrong.
2.2: Jaumann stress rate does not need superscript e.
Eq. (12): How do you calculate d_i? Please provide equation for it. It appears as scalar in Eq. (12), but difference between two points in stress space should be a vector.
Round 2
Reviewer 3 Report
Authors have addressed major comments satisfactory. Few minor things
1) Can you please note in section 2.3. that detailed description of DAMASK constitutive equations can be find in Roters et al 2019.
2) In new description of models, can you use following sentence: "The models are divided into: Model 1, Model 2 and Model 3."
